# Solar Powered Microplasma-Generated Ozone: Assessment of a Novel Point-of-Use Drinking Water Treatment Method

**DOI:** 10.3390/ijerph17061858

**Published:** 2020-03-13

**Authors:** Samuel Dorevitch, Kendall Anderson, Abhilasha Shrestha, Dorothy Wright, Aloyce Odhiambo, Jared Oremo, Ira Heimler

**Affiliations:** 1Division of Environmental and Occupational Health Sciences, University of Illinois at Chicago School of Public Health, Chicago, IL 60612, USA; Kendall.Anderson@ChicagoParkDistrict.com (K.A.); ashres2@uic.edu (A.S.); iheiml2@uic.edu (I.H.); 2Institute of Environmental Science and Policy, University of Illinois at Chicago, Chicago, IL 60612, USA; 3Presently employed by the Chicago Park District, Chicago, IL 60646, USA; 4Department of Biological Sciences, College of Liberal Arts and Sciences, University of Illinois at Chicago, Chicago, IL 60607, USA; dwrigh1@uic.edu; 5The Safe Water and AIDS Project (SWAP), P.O. Box 3323, 40100 Kisumu, Kenya; alloyce@swapkenya.org (A.O.); jared@swapkenya.org (J.O.)

**Keywords:** point-of-use drinking water treatment, household water treatment, ozone, microplasma, indicator bacteria, coliphage

## Abstract

Ozonation is widely used in high-income countries for water disinfection in centralized treatment facilities. New microplasma technology has reduced the energy requirements for ozone generation dramatically, such that a 15-watt solar panel is sufficient to produce small quantities of ozone. This technology has not been used previously for point-of-use drinking water treatment. We conducted a series of assessments of this technology, both in the laboratory and in homes of residents of a village in western Kenya, to estimate system efficacy and to determine if the solar-powered point-of-use water ozonation system appears safe and acceptable to end-users. In the laboratory, two hours of point-of-use ozonation reduced *E. coli* in 120 L of wastewater by a mean (standard deviation) of 2.3 (0.84) log-orders of magnitude and F+ coliphage by 1.54 (0.72). Based on laboratory efficacy, 10 families in Western Kenya used the system to treat 20 L of household stored water for two hours on a daily basis for eight weeks. Household stored water *E. coli* concentrations of >1000 most probable number (MPN)/100 mL were reduced by 1.56 (0.96) log removal value (LRV). No participants experienced symptoms of respiratory or mucous membrane irritation. Focus group research indicated that families who used the system for eight weeks had very favorable perceptions of the system, in part because it allowed them to charge mobile phones. Drinking water ozonation using microplasma technology may be a sustainable point-of-use treatment method, although system optimization and evaluations in other settings would be needed.

## 1. Introduction

In low- and middle-sociodemographic index (SDI) countries in 2017, an estimated 347 million cases of diarrheal illness among children under the age of five years attributable to unsafe source water occurred, resulting in 222,457 deaths [1]. For that reason, providing safe drinking water to populations in low- and middle-SDI countries is a high-priority global health challenge. The capital costs for centralized water treatment facilities and their maintenance greatly exceed the financial capabilities of low-income countries [2]. An estimated $17 billion per year (2.7% of Africa’s gross domestic product) would be needed annually to meet the drinking water development goals for Africa, with 43% of that allocated for new water infrastructure [3]. Less costly point-of-use (POU) methods for treating household stored drinking water have been promoted, including home chlorination, filtration, or by placing polyethylene terephthalate bottles containing water in sunlight for several hours (“solar disinfection” or SODIS) [4]. However, these approaches can be difficult to maintain, and are often abandoned by users because of the taste of chlorinated water, the time required to treat the water, and the ongoing need to purchase water treatment materials [5,6,7,8]. Furthermore, the evidence supporting health benefits (such as reductions in rates of childhood diarrhea) of POU treatment has been disappointing, due in part to limited adherence to POU treatment [9], which itself is impacted by factors such as the taste of treated water and complexity of the treatment [7,10].

Ozone (O_3_) is a potent disinfectant itself, and in water it promotes the formation of the hydroxyl radical. In the process of oxidizing substances in water, O_3_ is reduced back to molecular oxygen (O_2_). O_3_ has been used to disinfect wastewater and drinking water at large centralized facilities in high-income countries, particularly in Europe, for decades, and its use has grown rapidly [11]. A recent review of water disinfection technologies noted that while ozone is effective against bacteria, viruses, and protozoa (including *Cryptosporidium* spp.), O_3_ generation requires high-energy input and highly skilled technicians for system operation and maintenance [12]. 

Despite favorable features of drinking water ozonation, its reach in low- and middle-SDI countries is minimal. This is because conventional ozone production methods require reliable kilovolt-strength electrical fields and typically 100% oxygen as a feed gas (rather than ambient air) to produce O_3_ [13]. The World Health Organization notes that “ozone is not recommended for household water treatment because of the need for a reliable source of electricity to generate it, its complexity of generation and proper dosing in a small application, and its relatively high cost” [14]. 

In recent years, microplasma technology has been shown to dramatically reduce the strength of the electrical fields required for ozone generation [15,16,17]. A single chip (microchannel array) can produce 130 grams/kilowatt-hour, about three times more efficient than conventional dielectric barrier discharge (DBD) or corona ozone reactors that have comparable O_3_ outputs [17,18]. Microplasma units the size of a book can now generate O_3_ from the O_2_ present in ambient air, even in conditions of high relative humidity. Whether POU ozonation (POU-O_3_) may be a useful tool in the toolbox of POU methods is not known. In this initial assessment of the potential POU-O_3_, we evaluated its laboratory efficacy, field efficacy in controlled settings, household efficacy, and the user experience. 

## 2. Materials and Methods 

### 2.1. Study Elements Overview

The initial evaluation of the technology was conducted in Chicago, and addressed POU-O_3_ treatment of wastewater and its impacts on bacterial and viral indicators. Based on those results, experiments were conducted in Western Kenya that compared POU-O_3_ and POU chlorination of surface water. Next, 10 households in a Kenyan village used POU-O_3_ treatment of household-stored water over an 8-week period. We evaluated water quality impacts, the user experience, and safety considerations in the household study. 

### 2.2. Materials and Laboratory Methods

Ozone generators (EP Purelife 1000) were purchased from EP Purification (Champaign, IL, USA). The O_3_ output of the units is 0.2–0.3 gm/hour (depending on relative humidity), and air/ozone flow rates are approximately 2 L/min [17]. The ozone generators are 5.4 × 11.7 × 18.4 cm in size, and weigh 780 gm. The units have a single on/off button that turns on an internal pump that draws in ambient air, which flows through 250 µm in diameter channels in an aluminum block (reaction volume of 1.9 µL/channel). There, molecular oxygen (O_2_) in ambient air is converted to O_3_ in the plasma. The air/ozone mixture flows out of the ozone generator through Teflon® tubing and then through a ceramic aerator (diffuser) submerged in a vessel containing the water to be disinfected. The treatment system as deployed in Western Kenya is depicted schematically in Figure 1. Laboratory methods and instruments used are noted in Table 1 and described in Section 2.3 and Section 2.4. 

### 2.3. POU-O_3_ Impacts on Fecal Indicators in Wastewater, Chicago

Before considering the treatment of drinking water in Kenya, an initial assessment was conducted in the US to determine whether POU-O_3_ results in significant improvements in water quality. As levels of fecal indicators in Chicago area surface waters are often far lower than in surface waters used as a drinking water source in many parts of Western Kenya, we evaluated POU-O_3_ impacts on wastewater. While POU treatments of drinking water in Kenya generally involve 20 L volumes, to be conservative, we challenged the POU-O_3_ system by evaluating its impact on 120 L volumes of wastewater. Final effluent samples (40 L) were obtained from a wastewater treatment facility in Illinois that uses activated sludge treatment without disinfection. At the University of Illinois at Chicago (UIC) School of Public Health Water Microbiology Laboratory, wastewater was diluted with 80 L of de-ionized water in a plastic drum to bring *E. coli* concentrations to the 1000–2000 most probable number (MPN)/100 mL range that has been observed in surface waters in Kisumu, Kenya [21]. The water was sampled for analysis of three fecal indicators at baseline, and at 15, 30, 60, 90, 120, 150, and 180 min after ozonation we began using the methods noted in Table 1. At UIC, the ozone generators were powered using 110 V alternating current (not with solar panels). Production of ozone was verified with an aqueous ozone monitoring. The *E. coli* culture method with chromogenic substrate is considered by the USEPA to be comparable to membrane filtration for a variety of regulatory purposes [22]. The enterococci quantitative polymerase chain reaction (qPCR) method is used for beach water monitoring, as we have previously described [23]. The simplified rapid coliphage method (EasyPhage, Scientific Methods, Inc., Granger, IN, USA) was compared to EPA Method 1602 using seven serial dilutions of MS2 phage (obtained from Scientific Methods, Inc., Granger, IN, USA). These serial dilutions were split into six samples and assigned random ID numbers; three samples of each dilution were analyzed with the EasyPhage method in our laboratory and three samples were analyzed using EPA Method 1602 by Scientific Methods, Inc., a certified commercial laboratory. The lowest quantifiable measure of F+ coliphage was 6.25 plaque-forming unit (PFU)/100 mL lower limit of quantification (LLOQ), non-detects were assigned half that or 3.125 PFU/100 mL. To generate a preliminary assessment of the likelihood that POU-O_3_ may be able to reduce the concentration of indicator microbes in very turbid water (such as surface waters used as a drinking water source in many low- and middle-income countries), 12 additional trials of ozonation were conducted, in which organic matter was added to the water prior to treatment. Varying quantities of humic acid (Sigma Aldrich product 53680–50 G) were added to 100 L volumes of diluted wastewater in a plastic drum and stirred vigorously. In two trials, 3 grams of humic acid were added, and in 6 trials, a mean of 14 grams were added. The water was sampled for fecal indicator microbe analysis at baseline, and at 60 min after ozonation began. 

### 2.4. Laboratory Comparison of PO3 vs. POU Chlorine Treatment of Surface Water, Kenya

On 12 occasions between June 19 and July 22 2017, 60 L samples of surface water were collected from the River Kisian (approximately 0°05′03.1″ S, 34°40′43.0″ E) in Kisian Village, Kisumu County, Kenya. After thoroughly mixing the 60 L of collected water sample, 20 L water sample volumes were transferred into three clay pots. Water in one clay pot was treated with POU-O_3_ powered by a 15 W solar panel. Water in a second clay pot was chlorinated with 5 mL of 1.25% sodium hypochlorite per 20 L. Water in the third clay pot served as an untreated control. Immediately following the completion of treatment (ozonation: 120 min, and chlorine 5 min after mixing) and 1, 2, 4, 8, and 24 hours post-treatment, water was sampled for turbidity analyses and for *E. coli* culture. Prior to culture, water samples were diluted 10-fold, resulting in *E. coli* upper and lower limits of quantification of 24,197 and 10 MPN/100 mL, respectively. Samples treated with hypochlorite were dechlorinated shortly after sample collection with 1 mL of 3% sodium thiosulfate added per 100 mL of water samples. *E. coli* testing using the Colilert® method was conducted at the water quality laboratory of the Safe Water and AIDS Project (SWAP) in Kisumu, Kenya. 

### 2.5. Household Water Treatment with POU-O_3_, Kisian, Kenya

#### 2.5.1. Setting

In Kisian Village, Kisumu County, Kenya, few homes have access to either piped water or the electrical grid. Families generally obtain untreated surface water in 20 L jerry cans from the River Kisian, where cattle and animal fecal matter are often present. Others purchase river water from local entrepreneurs who deliver 20 L jerry cans via bicycle or motorbike. In the home, jerry cans are stored (or their contents transferred into 60–100 L “superdrums”) for settling for periods of hours to days. At that point, household stored water to be used for drinking (as opposed to washing) is transferred to a 20 L clay pot. Before or after transfer to the clay pot, household members might treat water using POU methods, most commonly, with chlorination [24]. 

#### 2.5.2. Household Enrollment and Intervention

Research team members from the SWAP, Kisumu, Kenya enrolled study participants. The eligibility criteria were (1) the presence of two or more children under the age of 5 years in the household, and (2) the use of surface water for drinking water. Two to three eligible households were recruited from each quadrant of the village for a total of ten participating households in Kisian. Participants were enrolled after completing a written informed consent process. Participants were asked to add POU-O_3_ to any water treatment method they routinely use, rather than replacing their current household water management practices with ozonation. Photovoltaic solar panels (15 W) were placed on the roofs of homes (Figure 1) and connected by an electrical cable to a POU-O_3_ unit. In addition to the port for powering the POU-O_3_ unit, the solar panels also had a USB port that could be used to charge mobile phones. A light bulb with an internal battery (essentially, a flashlight) was also provided, which could be charged during the day (when the POU-O_3_ system was not in use), and at night used for household lighting. Participants were asked to run the POU-O_3_ for two back-to-back 60-minute cycles each time they filled the 20 L clay pot. Participants were given the contact information of the SWAP researchers so that problems with the POU-O_3_ or other concerns could be communicated promptly. 

#### 2.5.3. Quantitative Data Collection

At weeks 0 (baseline), 1, 2, 3, 4, and 8 after the intervention began, computer-assisted interviews were conducted in Dhuluo using SurveyCTO software (Dobility, Inc., Cambridge, MA, USA) on Samsung tablet computers. The follow-up interviews focused on difficulties using the POU-O_3_ system. SWAP researcher staff sampled household stored water (HSW) and water from the clay pot that the family intended to use that day for drinking water (DW) that day (which were meant to have been treated by POU-O_3_). Information was also collected about the source of HSW (such as surface water, ground water, rooftop-harvested rainwater) and water samples were collected from those sources. The specific dates of home follow-up visits were unannounced, so that ‘real-life’ household water quality and treatment practices could be evaluated. HSW and DW samples were tested for turbidity and *E. coli* culture at the SWAP laboratory without dilution, resulting in an upper limit of *E. coli* quantification of 2419.7 MPN/100 mL. 

#### 2.5.4. Household POU-O_3_: Safety and User Experience in Kisian, Kenya

Eight weeks after household POU-O_3_ treatment began, two focus group discussions were conducted, one for male heads of households and one for female heads of households. The focus group discussion guide addressed perceptions of water quality and the use of POU-O_3_. The discussions were conducted in Dhuluo, recorded, transcribed, and translated into English for analysis. 

### 2.6. Quality Assurance

EPA reference methods for coliphage and enterococci qPCR testing include quality assurance processes and quality control samples. These were followed, including (for enterococci qPCR testing) standard curves, method blanks, salmon DNA as sample processing controls, and calibrators samples. The simplified coliphage testing performed at UIC included the use of coliphage positive (MS2 phage) and negative controls, and samples were read by two independent readers. At the SWAP laboratory in Kisumu, Kenya, turbidity meters were calibrated daily. Two independent analysts measured turbidity (in triplicate) and read *E. coli* results. 

### 2.7. Data Analysis

Descriptive statistics (mean, normality, standard deviation, median, 10th and 90th percentile, and range) were summarized for all water quality measures. To summarize the impacts of ozonation on wastewater (Chicago), surface water (Kenya) and household stored water (Kenya), the log removal value (LRV) was calculated as log10 (initial concentration/final concentration). The data were analyzed using SAS version 9.4 (SAS Institute, Cary, NC, USA) and graphing was done in Excel® (Microsoft, Redmond, WA, USA). Focus group data were analyzed using Atlas ti® software through a process of coding and theme construction [25].

### 2.8. Human Subjects Research Protections

The study of household use and user perceptions of POU-O_3_ was conducted following a research protocol approved by the University of Illinois at Chicago (UIC) IRB (Protocol 2017-0572) and the Maseno University Ethics Review Committee (MUERC), in Kisumu, Kenya (Protocol 00419/17). 

## 3. Results

### 3.1. Wastewater POU-O_3_ Treatment, Chicago

Water quality measures at baseline approximated a normal distribution and are summarized in Table 2. At baseline, O_3_ concentrations in wastewater samples were <0.02 ppm, and during ozonation they plateaued at 0.28–0.40 ppm after 45 min of treatment. Changes in fecal indicator microbes during ozonation are presented as LRV in Figure 2. Mean (standard deviation) LRV of *E. coli* at 120 min was 2.34 (0.84) for enterococci qPCR 1.16 (0.72) and further LRV reduction for enterococci were observed for enterococci over the subsequent 60 min. The LRV for F+ coliphage at 60 min of ozonation was 1.54 (0.43). The simplified coliphage method produced results strongly correlated with those of EPA Method 1602 (R^2^ = 0.99). EasyPhage results are consistently approximately 2/3 the numerical value of the EPA method (Appendix A and images of EasyPhage plaque forming units in Appendix A).

The addition of humic acid to the 100 L samples of diluted wastewater prior to ozonation reduced the impact of ozonation on culturable *E. coli* and on qPCR measures of enterococci, but had no discernable impact on the effectiveness of ozone on F+ coliphage (Appendix A).

### 3.2. Comparison of POU-O_3_ and POU-Chlorine Treatment of Surface Water, Kenya

At baseline, River Kisian *E. coli* mean (standard deviation) concentration (*n* = 11) was 4426.4 MPN/100 mL (3649.8) and turbidity (*n* = 10) was 109.0 NTU (56.1). During the trials, mean (standard deviation) relative humidity, which impedes conventional ozone generation, was 64.8 (6.7) percent. The control samples (untreated) showed no change in *E. coli* (LRV −0.02 (0.13) during the time that other samples were ozonated. Immediately after chlorination, *E. coli* concentrations were reduced by mean (standard deviation) LRV 3.43 (0.46). While the overall LRV for ozonation trials was 1.56 (1.34), in the early trials the treatment system was found to have mechanical problems, with field notes describing leaky connections between the ozone generator and ozone-out tubing, absence of bubbles in the water during treatment, “no ozone smell present,” or “treatment system moved during treatment.” Subsequently, methods for connecting the ozone generator and the tubing were improved. Thus, two distinct populations of results were obtained: in five trials with mechanical problems the LRV was 0.11 (0.10), while in the other six trials (in which simple mechanical or other problems were not encountered) it was 2.43 (0.86). 

### 3.3. Household POU-O_3_, Kisian, Kenya

Fifty follow-up visits occurred (10 households at five time points), but 46 pairs of *E. coli* results (HSW, DW) were available (92%) because either HSW or DW was not available for sampling on four home visits. Water quality is summarized in Table 3. 

LRV describing differences in paired measures of HSW and DW was highly variable. Overall, the mean (standard deviation) LRV was 1.1 (0.86). Large reductions in LRV are not possible if the baseline (HSW) *E. coli* concentrations are low. Among the 15 observations of HSW *E. coli* ≥ 1000 MPN/100 mL, LRV was 1.56 (0.96). As the measured *E. coli* was truncated at the upper limit of quantification (2419.7/100 mL) for 12 of these 15 observations, the reported LRV uderestimates the actual LRV, although the degree of underestimation is unknown. Among the 12 observations of HSW *E. coli* 100–999 MPN/100 mL, LRV was 1.37 (0.73), and in 19 trials with HSW *E. coli* <100, LRV was 0.59 (0.57). Of the 21 households with HSW samples >10^3^
*E.coli* MPN/100 L, 16 (76%) had DW *E. coli* of 10^0^ MPN/100 mL. High LRV values were not limited to specific households: on seven occasions involving six different households, LRV was between 2.3 and 3.3. No linear trend in LRV over the five rounds of follow-up was observed. 

### 3.4. User Perceptions of the Ozonation System, Kisian Kenya

Nine of 10 participants denied having any problems using the POU-O_3_ at all rounds of follow-up. One participant noted on one occasion that she could hear that the treatment system was humming (as expected) during operation, but bubbles were not produced (indicating a problem with the connection between the tubing and the ozone generator). The connection was tightened and additional education and training was provided. 

The focus group discussion for female heads of household had participation of all 10 households; the focus group discussion for male heads of household had six participants. Both groups expressed enthusiasm for the POU-O_3_ system, due to the perceived benefits of the water treatment on water quality, as well as the ability to charge mobile phones and other devices from the solar panel’s USB port. Among reasons given for favorable impressions of the system was the fact that bubbles of ozone-enriched air produced in the clay pot during treatment gave the appearance of boiling, which is viewed as a very good way to disinfect water; many of the participants referred to the process as “boiling water”, even though they knew that the temperature of the water did not increase during treatment. Several participants noted that they liked the taste of the treated water, and there were anecdotal mentions of decreases in abdominal cramping since POU-O_3_ began. Several participants reported that neighbors brought jerry cans of water to their homes to have their water treated as well. Several but not all participants commented on the smell of ozone when using POU-O_3_ was in use, but that the water itself did not develop a similar taste. Although the odor of ozone was detectable during (but not immediately after) water treatment, when asked, participants denied that family members had symptoms of respiratory or eye irritation during operation of the POU-O_3_ system. All participants asked if they could keep and continue using the water treatment systems after the study was completed. All were given the devices to keep and were noted months later to still use the devices. Approximately one year later, one solar panel disappeared from a home, the cooling fan of a POU-O_3_ unit required replacement, and the pump of another unit required replacement. The smell of ozone during operation continued as expected, indicating that the microplasma unit continued to function. 

## 4. Discussion

Plasma, a gas of ions, is one of the four states of matter, along with solid, liquid, and gas. The ability to generate a plasma on a small scale has created the possibility of low-energy, small-scale ozone generation for POU water treatment. In this initial evaluation of microplasma generated POU-O_3_, we found that the method can significantly reduce concentrations of fecal indicators. In ‘real-world’ settings of a Kenyan village where source water had very high turbidity, concentrations of *E. coli* were reduced substantially, although not consistently brought to undetectable levels. Users of the solar-powered intervention embraced it enthusiastically, and denied symptoms of mucous membrane and respiratory irritation that could result from O_3_ exposure. 

### 4.1. Evaluation of Microplasma-Generated O_3_ for POU Water Treatment

The World Health Organization (WHO) has developed and applied a framework for evaluating household water treatment methods [26,27]. Using that framework, the WHO reviews independently-produced data and/or commissions independent testing of treatment methods to determine LRVs for *E. coli*, coliphage virus, and *Cryptosporidium* spp. Based on the data presented here (which do not include protozoa testing), the POU-O_3_ system might be categorized intermediate between “2 stars” and “1 star” based on 2 LRV for *E. coli* reduction. Coliphage reduction of <2 LRV was observed, although this was for 120 L treatment volumes. The real-world performance in homes of users—which is not required under the WHO framework—demonstrated 2 LRV reduction in *E. coli,* despite the fact that median turbidity of 49.7 NTU exceeded the 40 “challenge” level of the WHO evaluation framework. The impact of POU-O_3_ on water quality that we observed in the homes of village residents was comparable to POU filtration, chlorination, and combined chlorination/flocculation reported from a study conducted elsewhere in Kenya [6].

An alternative approach to evaluating POU water treatment that includes additional considerations has been described [28]. That framework assesses performance (LRV of fecal indicators), ease of use, water throughput, acceptability potential, energy requirement, cost, ease of deployment, durability, maintenance, environmental impact, and supply chain factors. The present study was not designed to evaluate all of these parameters; however, the ozonation system would be assed favorably in terms of acceptability (based on focus group discussions), energy requirements, environmental impact, ease use, and supply chain. User acceptability—beyond ease of use—was high in part because of the production of bubbles (“like boiling”) during water treatment, as well as because of the opportunity to charge mobile phones using the solar panel’s USB port. These factors may have contributed to observations that the reach of the intervention extended beyond the households enrolled and to neighbors who brought their HSW to homes of participants for treatment. 

As deployed in Kisian, throughput was 20 L/2 hours, and additional rounds of treatment could certainly be conducted on the same day. Regarding ease of deployment, a solar panel could simply be placed on the ground, though we secured the panels on roofs of participants’ homes, which required relatively little effort. We purchased the microplasma ozone generators for $250, which would be far beyond what residents of low-income communities could afford. By comparison, the cost for nine months of POU flocculation/chlorination in Pakistan has been estimated to be approximately $100 per child [29], though POU chlorination alone (without flocculation) has been estimated to cost less than $10 per year [30]. While the current cost of POU-O_3_ is far greater than chlorination, the POU-O_3_ system is modular and scalable. Thus, an array of solar panels and ozone generators should be able treat far larger volumes of water in a community-scale decentralized system, bringing down the cost of water on a per m^3^ basis. 

### 4.2. Limitations of POU-O_3_

We note that a limitation of the system is that before the system could be more widely deployed, connections between the ozone generator and the “ozone out” tubing must be made more secure, which we did after the controlled trial and before deployment in Kisian homes. Both the use of solar power and ozone have limitations. Many of the countries that lack access to basic drinking water are the same countries with the most solar irradiation (due to their equatorial locations and climates) [31], making photovoltaic solar technologies potentially useful; however, the climate in other locations may not support the consistent availability of solar power. Unlike chlorination, ozonation does not produce a residual that persists during water storage, making the water susceptible to re-contamination. For that reason, narrow mouth storage containers should be used for water treatment; this would also ensure that ozone generators do not accidentally fall into the water. 

Ozone can oxidize many plastics, as well as metals found in ground water, resulting in rust colored treated water. Bromide-containing compounds may be present in groundwater, which when oxidized can produce bromate [32,33]. This is a concern because potassium bromate is a possible human carcinogen [34]. Thus, the assessment of bromine concentrations in groundwater may be needed before ozonation is considered as a treatment option. In such settings, the potential benefits of reducing the risk of acute gastrointestinal infections may have to be weighed against increases in potential long-term chronic health risks. Ozonation and other oxidation processes can cause the degradation of selected pesticides, the products of which, when chlorinated, can generate chlorinated byproducts that exhibit toxicity in in vitro assays [35]. Thus, the use of oxidation processes followed by chlorination may not be appropriate for the production of drinking water from pesticide-containing source waters.

### 4.3. Limitations and Strengths of this Study

This initial evaluation of POU-O_3_ is limited by the number of trials of ozonation in the laboratory in Chicago (four trials of 100 L disinfection), in controlled trials of POU-O_3_ vs. chlorination in Kenya (11 trials of 20 L disinfection), and in the setting of household use (10 households followed for 8 weeks). The standard deviations of LRV in the Chicago trials of wastewater disinfection are large in relation to the mean. The raw data from the small number of individual trials do not contain an obvious outlier, suggesting that the trials contained significant “noise” due to variability in ozone output, microbe viability in wastewater, adherence to the laboratory protocol, and/or measurement error in quantifying microbe concentrations. Calibrator sample and standard curve data of the qPCR system do not suggest measurement error. The same is true of the EasyPhage comparison with EPA Method 1602. Greater control of all conditions will be needed in future studies to identify the source(s) of variability. Thus, the precision of LRV estimates and the range of user perspectives were limited. Likewise, a longer duration of follow-up in the household use study might have identified decreases in LRV over time. A strength of the study is that the research team did not announce their follow-up visits to households in Kisian, and, as a result, the HSW and DW samples collected reflected real-world conditions of use (or non-use). This study of POU-O_3_ use by study participants in Kisian did not include a comparison group that used chlorine for disinfection, and for that reason, the relative performance of the two POU methods is unknown. Participants denied the development of respiratory symptoms and those of mucous membrane irritation. It should be noted that participants lived in very well-ventilated homes; use in “tight” buildings would not be recommended without ensuring that ozone levels remain below the threshold of respiratory irritation. 

### 4.4. Other Findings

Our finding that ozonation of wastewater in the laboratory setting resulted in greater and more rapid reductions in concentrations of culturable bacteria than qPCR measures of bacteria is consistent with the findings of Sousa et al., who also noted larger relative reductions in concentrations of several fecal indicator bacteria measured by culture, relative to qPCR measures of bacterial targets following ozonation (generated using conventional methods) [36]. This may be because unlike culture-based methods, the qPCR method does not differentiate between viable, viable but non-culturable, and non-viable bacteria. Starting concentrations of F+ coliphage in wastewater were lower than the bacterial indicators, but they responded more rapidly to ozone than the bacteria did, even in the presence of humic acid. This is consistent with studies (which did not involve the on-site generation of ozone) that described large reductions in viral pathogens and/or indicators following relatively brief ozonation [37,38,39]. 

### 4.5. Research Needs

Further research might include the household use of PO_3_ in other settings with larger numbers of participants. We found in the controlled trials of turbid surface water disinfection in Kenya that the dose of chlorine used was superior—in terms of *E. coli* reduction—than the dose of ozone used. Future work might address optimal ozone (gm/liter and treatment time) over a range of turbidity values. PO3 effects on protozoa, costs, impacts on the occurrence of diarrheal disease in children, and user acceptability in other cultures should also be assessed. Additionally, identifying methods to reduce the cost per m^3^ of water produced might expand the population that could benefit from this approach to water treatment. Efforts to ensure the safe water storage of ozonated water would also be important, given the absence of a disinfectant residual with O_3_ treatment. Direct comparison of the disinfection effects of POU-O_3_ and chlorine, as well as comparisons of health impacts, would be useful. 

## 5. Conclusions

This first assessment of a point-of-use ozonation device that uses microplasma technology for drinking water treatment shows that the method holds promise for addressing global water challenges in low- and middle-income countries. We found that it can significantly lower concentrations of *E. coli* and coliphage virus in 120 L of wastewater following 60 min of treatment. The method was able to reduce *E. coli* concentrations substantially in heavily contaminated, turbid surface water in 10 households in Kenya over 8 weeks of follow-up. The point-of-use method was readily adopted and viewed favorably by families. Much more work would be needed to determine what role this method might play in addressing global drinking water quality challenges.

## Figures and Tables

**Figure 1 ijerph-17-01858-f001:**
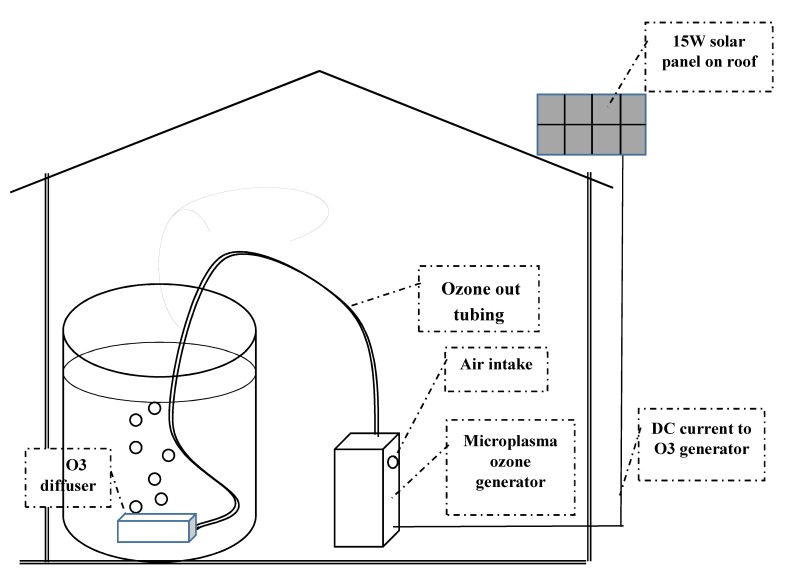
Schematic diagram of the point-of-use ozonation (POU-O_3_) system as deployed in Kenya. At the University of Illinois at Chicago (UIC), the system was operated using 110 V alternating current, rather than direct current from a solar panel.

**Figure 2 ijerph-17-01858-f002:**
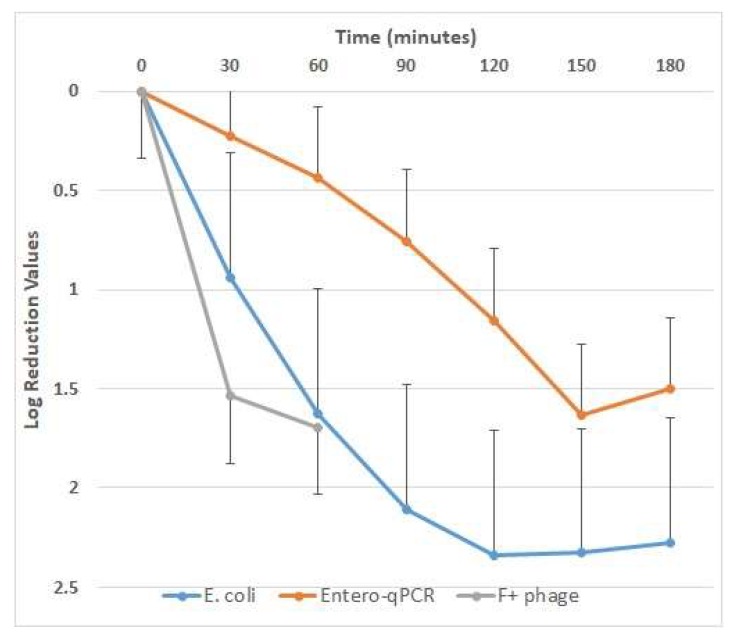
Changes in indicator microbe concentration in wastewater, as log removal value (LRV) following ozonation. Error bars (up-going or down-going) indicate standard deviation across all trials for each indicator microbe.

**Table 1 ijerph-17-01858-t001:** Water analysis methods and instruments.

	Used In	Laboratory Method and/or Instrument
Aqueous ozone	Chicago	ATI Q46A Dissolved Ozone Monitor (Analytical Technology, Inc, Collegeville, PA, USA).
pH	Chicago	Orion™ Dual Star, Thermo Scientific (Waltham, MA, USA)
Turbidity	Chicago, Kenya	LaMotte 2020we (Chestertown, MD, USA)
*E. coli*	Chicago, Kenya	Defined substrate culture (Colilert®), IDEXX Laboratories, Westbrook, ME, USA
Enterococci	Chicago	qPCR, USEPA Method 1609.1 [19] QuantStudios 3 Thermocylcer, ThermoFischer
F+ coliphage	Chicago	Rapid coliphage (“EasyPhage”), Scientific Methods, Inc.
F+ coliphage	Scientific Methods, Inc., Granger, IN, USA	USEPA Method 1602 [20]

qPCR: quantitative polymerase chain reaction. USEPA: United States Environmental Protection Agency.

**Table 2 ijerph-17-01858-t002:** Characteristics of diluted wastewater (100 L) prior to ozonation at UIC, at baseline. *n* = 4 trials, except for F+ coliphage, which was measured in 3 trials.

Water Quality Parameter	Mean (Standard Deviation)	Range
pH	7.22 (0.24)	(7.02, 7.49)
Turbidity (NTU)	1.62 (0.89)	(1.02, 2.94)
*E. coli* (MPN/100 mL)	1501.0 (924.5)	(965.0, 2885.0)
Enterococci (CCE/100 mL)	11,059.7 (5659.4)	(3572.9, 16,627.6)
F+ coliphage (PFU/100 mL)	247.9 (41.6)	(212.5, 293.8)

**Table 3 ijerph-17-01858-t003:** Water quality in the homes of participants, Kisian Village. Units of measure for *E. coli*: MPN/100 mL. Units of measure for turbidity: NTU. *N* = 46.

Water Type	Water Quality Measure	Median	(10th, 90th Percentile)
Household stored water	*E. coli*	203.7	(7.9, 2419.7)
	Turbidity	48.61	(2.72, 430.3)
Drinking water	*E. coli*	11.4	(0.9, 369.7)
	Turbidity	40.23	(1.34, 373.0)

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
