# Peer review of "Solar Powered Microplasma-Generated Ozone: Assessment of a Novel Point-of-Use Drinking Water Treatment Method"

_ijerph, 2020, doi:10.3390/ijerph17061858_

Round 1
Reviewer 1 Report
Authors have performed a deactivation/disinfection of common bacteria’s by ozone. They have concluded that this method could be a sustainable point-of-use treatment method.
Developing solutions for drinking water treatment (especially in countries with a low sociodemographic index) is very important. Ozone is being used for water purification purposes for decades, however, research is still ongoing in order to e.g. further investigate its oxidizing capability for emerging micropollutants.
The study is written in good English and fits to the scope of the journal, however, in my opinion, the amount of data provided could be increased. Maybe additional analyses of the provided data could be conducted. Overall, many shortcomings are listed below and at least major revision should be conducted.
- Figure 1 is not generated correctly in my PDF (e.g. some descriptions “O3 generator” are not visible).
- 1 description: and in Kenya the power was generated by solar panels?
- Additional discussion about emerging topic - heterogeneous oxidation e.g. with ozone could be added basing on a recent review: https://doi.org/10.1515/eces-2018-0001
- and potential dangers of AOPs, I would suggest: Edyta Kudlek, Identification of Degradation By-Products of Selected Pesticides During Oxidation and Chlorination Processes, 2019.
- Ozone is a toxic gas and could be harmful for residents of a house in which this setup would be implemented. Authors have considered such possibility?
- What about the ozone concentrations in your study, were they measured?
- If you would implement some kinetic model in your study, it could improve manuscripts quality.
- 2 standard deviations are really large! Do you have some explanation why there is such deviation?
- What about the energy efficiency of your treatment?
- L53: O3 is reduced back to molecule – subscript.
- L58: “Unlike chlorination, ozonation of surface water does not produce potentially carcinogenic disinfection byproducts.” – I think that this could be not completely true, there are some special cases where the toxicity of the treated by ozone medium can be raised, e.g. https://www.sciencedirect.com/science/article/abs/pii/S004313540200458X
- L89: “Error! Reference source not found”
- L95 and further: Table caption should be under the table? Please check in guide for authors.
- L111: “instrumentsError! Reference source not found..”
- L206: “Error! Reference source not found..” – please check carefully manuscript for these errors.
- L245: “of these15 observations”
- L287: “microplasma-generated O3 for POU water” – subscript.
Author Response
Dear Reviewer 1,
Thank you for your helpful comments.
A detailed, item-by-item response to your comments, as well of those of other reviewers, is attached.
Best,
Sam Dorevitch

Reviewer 2 Report
-A mechanicistic study on the ozonization process would be useful. How the authors think that safe by-products have been produced? As chlorinated by-products, the ozone by-products may be toxic and dangerous
-Fig. 2. Please, correct the y-label. Which the meaning of F+phage trend during time, why was it evaluated untill 60 min of ozonization?
-Which the meaning of so large error bars in Fig. 2?
-Why the increasing trend of E-coli and Entero-qPCR after 150 min?
-A comparison of the costs of this process with conventional ones must be reported. The appliction of this techniques must be necessarily cheap. On the contrary, the addition of chlorine would be more efficient and less expensive.
-many typing errors as line 89, 111, 206 and 213 must be corrected.
-Please, insert the error bars also in the figure of the supplementary file.
Major revisions are required
Author Response
Dear Reviewer 2,
Thank you for your helpful comments.
A detailed, item-by-item response to your comments, as well of those of other reviewers, is attached.
Best,
Sam Dorevitch

Reviewer 3 Report
The Authors proposed a very interesting topic for research. The paper is publishable, but a major revision is required before the manuscript acceptance.
- The article contains a editorial errors e.g. line 53, 55, 61, 82, 85, 89, 97, 111, 100-138, 143, 206, 213, 238, 245, 292, 293, 332, 363, 371 etc.
- Figure 1 is illegible.
- The purpose of the work should be accurately described and a new aspect specified.
- The discussion of the results lacks a reference to the research of other scientists, a comparison with their results.
Author Response
Dear Reviewer 3,
Thank you for your helpful comments.
A detailed, item-by-item response to your comments, as well of those of other reviewers, is attached.
Best,
Sam Dorevitch

Round 2
Reviewer 1 Report
Authors have improved the manuscript, I would only suggest to control the number of significant figures in Table 2 (https://en.wikipedia.org/wiki/Significant_figures). For example, should you write 1,501.0 if you have a standard deviation of 924.5?
Reviewer 2 Report
The paper is now suitable for publication
Reviewer 3 Report
Thank you for responding to my suggestions. I recommend this article for publication.